# Chemical Characterization and Several Bioactivities of *Cladanthus mixtus* from Morocco

**DOI:** 10.3390/molecules28073196

**Published:** 2023-04-03

**Authors:** Amina El Mihyaoui, El Hadi Erbiai, Saoulajan Charfi, Eugénia Pinto, María Emilia Candela Castillo, Josefa Hernández-Ruiz, Antonio Cano, Alain Badoc, Ahmed Lamarti, Joaquim C. G. Esteves da Silva, Marino B. Arnao

**Affiliations:** 1Department of Plant Biology (Plant Physiology), Faculty of Biology, University of Murcia, 30100 Murcia, Spain; elmihyaoui.amina@gmail.com (A.E.M.); mcandela@um.es (M.E.C.C.); jhruiz@um.es (J.H.-R.);; 2Laboratory of Plant Biotechnology, Department of Biology, Faculty of Sciences, Abdelmalek Essaadi University, Tetouan 93000, Morocco; lamarti.ahmed58@gmail.com; 3Centro de Investigação em Química (CIQUP), Departamento de Geociências, Ambiente e Ordenamento do Território, Instituto de Ciências Moleculares (IMS), Faculdade de Ciências, Universidade do Porto, 4169-007 Porto, Portugal; elhadi.erbiai@etu.uae.ac.ma (E.H.E.);; 4Laboratory of Biology, Environment and Sustainable Development, Higher School of Teachers (ENS), Abdelmalek Essaadi University, Tetouan 93150, Morocco; 5Laboratory of Biology and Health, Department of Biology, Faculty of Sciences, Abdelmalek Essaadi University, Tetouan 93000, Morocco; sawlajan@gmail.com; 6Laboratory of Microbiology, Biological Sciences Department, Faculty of Pharmacy, University of Porto, 4050-313 Porto, Portugal; epinto@ff.up.pt; 7CIIMAR-Interdisciplinary Center of Marine and Environmental Research, University of Porto, 4450-208 Matosinhos, Portugal; 8INP, INRAE, OENO, UMR 1366, ISVV, Université de Bordeaux, Villenave d’Ornon, F-33140 Bordeaux, France; jbtalence@free.fr

**Keywords:** antioxidant activity, antibacterial activity, antifungal activity, *Aspergillus fumigatus*, *Cladanthus mixtus*, *Candida albicans*, *Escherichia coli*, medicinal-aromatic plants (MAPs), phytotherapy, *Staphylococcus aureus*, *Trichophyton rubrum*

## Abstract

The purpose of this work was to investigate, for the first time to our knowledge, the chemical composition and bioactivity of methanolic extracts (roots, stems, leaves, and flowers) from *Cladanthus mixtus* (L.) Chevall. that grows wild in northern Morocco (the Tangier-Tetouan-Al Hoceima region). The phenolic and flavonoid contents were determined by spectrophotometer methods, and the composition of derivatized methanolic extracts from *C. mixtus* using *N*-*O*-bis(trimethylsilyl) trifluoroacetamide (BSTFA) was analyzed by gas chromatography–mass spectrometry (GC-MS). The antioxidant activity was carried out by applying the 2,2′-azino-bis-(3-ethylbenzothiazoline-6-sulfonic acid) (ABTS) and DPPH (2,2-diphenyl-1-picrylhydrazyl) tests. The micro-dilution technique was chosen to investigate the antimicrobial activity of methanolic extracts against two bacterial strains and three fungal species. The results showed that the values of total phenolic and flavonoid contents were found to be higher in flower extracts (30.55 ± 0.85 mg of gallic acid equivalents (GAE)/g of dried weight (DW) and 26.00 ±1.34 mg of quercetin equivalents (QE)/g DW, respectively). Other groups of chemical compounds were revealed by GC-MS, such as carbohydrates (27.25–64.87%), fatty acids (1.58–9.08%), organic acids (11.81–18.82%), and amino acids (1.26–7.10%). Root and flower methanolic extracts showed the highest antioxidant activity using ABTS (39.49 mg of Trolox equivalents (TE)/g DW) and DPPH (36.23 mg TE/g DW), respectively. A positive correlation between antioxidant activity and polyphenol and flavonoid amounts was found. Antibacterial tests showed that the best activity was presented by the leaf extract against *Staphylococcus aureus* (minimum inhibitory concentration (MIC) = minimum bactericidal concentration (MBC) = 20 mg/mL) and *Escherichia coli* (MIC of 30 mg/mL and MBC of 35 mg/mL). *S. aureus* was more sensitive to the extracts compared to *E. coli*. All extracts showed antifungal activity against *Trichophyton rubrum*, with the best efficacy reported by the flower and leaf extracts (MIC = 1.25 mg/mL and minimum fungicidal concentration (MFC) = 2.5 mg/mL). In general, extracts of *C. mixtus* appeared less effective against *Candida albicans* and *Aspergillus fumigatus*.

## 1. Introduction

Plants are used in traditional medicine to treat a wide range of ailments. These medicinal-aromatic plants (MAPs) are well known for their biological activity. The World Health Organization estimates that over 80% of the global population still has confidence in conventional and folk medicine, mostly based on herbal remedies [1,2]. Herbs and plant-derived products have a long history of safe use as natural products in the treatment of various diseases [3].

Morocco is known for its beneficial geographical location as a country with Mediterranean and Atlantic coasts, which has contributed to an interesting plant diversity [4,5]. The Asteraceae family is the largest flowering herb family, with over 1700 genera and 34,000 species worldwide. It involves several plants with medicinal values, such as chamomile, wormwood, and dandelion, among others [6].

Studies have shown that some Asteraceae plants have many biological properties, such as antioxidant [7], antifungal [8], antibacterial [9], anti-inflammatory [10], and anticancer activities [11].

*Cladanthus mixtus* (L.) Chevall. (Moroccan chamomile or simple leaved chamomile, synonymous with *C. mixtus* (L.) Oberpr. and Vogt., *Anthemis mixta* L., *Chamaemelum mixtum* (L.) All., and *Ormenis mixta* subsp. *mixta*) belongs to the Asteraceae family [12]. This traditional medicinal plant is widespread in Morocco and in the northern and eastern zones of the Mediterranean basin. The flowers and leaves of *C. mixtus* are the most commonly used parts as an infusion to treat various diseases [13,14]. Furthermore, *C. mixtus* is used by therapists and herbalists as an antispasmodic, analgesic, antiallergic, anti-inflammatory, carminative, digestive, febrifuge, fungicide, vermifuge [15], antioxidant agent [16], and anticancer treatment [11].

In a previous study by El Mihyaoui et al. [11], the HPLC-MS analysis of methanolic extracts from *C. mixtus* revealed the presence of 23 phenolic compounds identified in the flowers and 24 compounds in the leaves, stems, and roots extracts; the GC-MS analysis of methanolic extracts without derivatization showed that *C. mixtus* is rich in biomolecules, including terpenoids, alcohols, esters, alkanes, fatty acids, organic acids, benzenes, phenols, ketones, sterols, carbonyls, amines, and other groups.

Plants can produce many diverse bioactive compounds, and several factors can affect the yield of these compounds, such as different extraction solvents and techniques as well as the particular isolation and purification of bioactive molecules [17].

The aim of this work is to investigate the chemical composition of methanolic extracts from different plant organs (flowers, leaves, stems, and roots) of wild Moroccan *C. mixtus* and to evaluate their biological activities, including antioxidant, antibacterial, and antifungal properties. To the best of our knowledge, this is the first comparative research study on the chemical characterizations and biological activities of different *C. mixtus* organs (flowers, leaves, stems, and roots) from northern Morocco (Tangier-Tetouan-Al Hoceima region).

## 2. Results

### 2.1. Extraction Yield and Total Polyphenol and Flavonoid Contents

The extraction yield was determined on 2 g of dry plant material and was expressed as a percentage. The results obtained are shown in Table 1. The flowers of *C. mixtus* gave the highest yield (25.86%). The roots also gave a good yield (20.65%), followed by leaves (19.40%), and stems (18.75%).

The results in Table 1 also illustrated the polyphenol and flavonoid contents of methanolic extracts of the different organs of *C. mixtus*. The polyphenol contents ranged from 16.43 to 30.55 mg GAE/g DW. The content was significantly higher in flowers (*p* < 0.05). Roots and stems showed almost the same content (18.83 and 18.77 mg GAE/g DW, respectively), while the lowest content was observed in leaves (16.43 mg GAE/g DW). Concerning flavonoids, the contents ranged from 8.74 to 26.00 mg QE/g DW, following the order: flowers > stems > leaves > roots. Consequently, flowers showed the highest content of polyphenols and flavonoids.

### 2.2. Biochemical Constituents of Cladanthus mixtus Organs by GC-MS

GC-MS chromatograms of derivatized extracts from flowers, leaves, stems, and roots of *C. mixtus* at different retention times (Figure 1, Figure 2, Figure 3 and Figure 4) revealed the presence of 42, 74, 70, and 83 phytochemical compounds, respectively. These compounds can be mainly divided into six groups, including carbohydrates, lactones, organic acids, fatty acids, phenols, amino acids, and other biomolecule groups. In general, the extracts from flowers, leaves, stems, and roots were dominated by sugars (27.25, 54.8, 64.87, and 62.57%, respectively) (Table 2). Sucrose was observed to be the main biomolecule detected in roots (22.47%), flowers (17.01%), and stems (14.84%), while myo-inositol was the major compound identified in leaves (9.38%).

As presented in Table 3, the results of derivatized methanolic extracts from *C. mixtus* showed that carbohydrates were important and varied. Flower extracts contained 10 compounds, representing 27.25% of the total biomolecules detected, and sucrose (17.01%) was the main constituent identified. In contrast, 20 biomolecules were identified in leaf extracts, representing 54.80% of leaf compounds, which were dominated by myo-inositol (9.38%), meso-erythritol (6.60%), and glucose (6.39%). In stem and root extracts, 28 and 27 compounds were detected, representing 64.87% and 62.57% of compounds, with sucrose (14.84% and 22.47%), myo-inositol (13.9% and 6.65%), and D-fructofuranose (7.37% and 8.46%) as the major carbohydrates in both organ extracts, respectively.

Concerning lactones, the GC-MS analysis showed their presence only in flower extracts (Table 4). The extract contained 2 compounds: D-glucurono-γ-lactone (24.26%) and erythrono-1,4-lactone (0.34%).

Following carbohydrates, organic acids represented an important percentage of biomolecules identified in *C. mixtus* derivatized extracts, whereas 11.81%, 18.82%, 17.66%, and 13.00% of organic acids were observed in flowers (5 compounds), leaves (10 compounds), stems (8 compounds), and roots (11 compounds), respectively (Table 5). Malic acid was the major organic acid detected in the four organ extracts.

Regarding fatty acids (Table 6), 2 molecules were detected in the flower and stem extracts, with linoelaidic acid (6.73%) as the main compound in the flowers and dimethyl malate (1.05%) in the stems. Leaf and root extracts contained 7 and 6 compounds, respectively, with docosahexaenoic acid (2.71%) as a major component in leaves and palmitic acid (1.83%) in roots.

For phenolics, the GC-MS analysis illustrated the presence of 2 biomolecules in flowers: chlorogenic acid (5.92%) and naringenin (1.16%). However, chlorogenic acid (1.5%) was the only molecule detected in the leaves. Otherwise, chlorogenic acid and caffeic acid were the phenolics identified in the stem (4.03% and 4.43%) and root (2.26% and 4.01%) extracts, respectively (Table 7).

Concerning amino acids, the results differed widely between the studied organs (Table 8), whereas no amino acids were detected in flower extracts, unlike the 6 compounds identified in leaves and 2 in each of the stem and root extracts, representing a total of 7.10%, 1.26%, and 2.28%, respectively.

For the rest of the biomolecules obtained by GC-MS (Table 9), many biological groups were detected, including terpenoids, alcohols, benzenoids, alkanes, and pyrimidines. In *C. mixtus* roots, 19 molecules (12.31%) were reported, while 14 molecules (21.13%) were reported in flowers, 14 molecules (8.69%) in leaves, and 11 molecules (5.15%) in stems.

### 2.3. Antioxidant Activity

The antioxidant activity of methanolic extracts from different organs of *C. mixtus* was determined by DPPH and ABTS methods, and the results are shown in Figure 5. The DPPH results ranged from 16.96 to 36.23 mg TE/g DW, with a significant difference between all organs (*p* < 0.05). Flowers showed the highest activity (36.23 mg TE/g DW), followed by roots (30.24 mg TE/g DW), stems (24.72 mg TE/g DW), and leaves (16.96 mg TE/g DW). For the antioxidant activity of *C. mixtus* extracts determined by the ABTS method, the results were significantly different (*p* < 0.05) between the four organs, with values ranging from 12.58 to 39.49 mg TE/g DW. The strongest antioxidant activity was obtained from roots (39.49 mg TE/g DW), followed by flowers (33.6 mg TE/g DW), stems (20.99 mg TE/g DW), and leaves (12.58 mg TE/g DW).

### 2.4. Antibacterial Activity

The extracts of the four organs showed antibacterial activity in vitro against the Gram-positive and Gram-negative bacterial strains *S. aureus* and *E. coli* (Table 10). *S. aureus* was most sensitive to the leaf and flower extracts (MIC = MBC = 20 and 32 mg/mL, respectively). Moreover, the stem and root extracts were also active against *S. aureus,* with MIC and MBC values of 40 mg/mL. For *E. coli*, the leaf extracts showed the best activity, with an MIC value of 30 mg/mL and MBC of 35 mg/mL. The other extracts from the roots, stems, and flowers showed activity against *E. coli* with MIC and MBC values of 40 mg/mL. Between the two strains tested, *S. aureus* was more sensitive to the extracts.

When comparing the antibacterial activity of the extracts with that of a reference antibiotic (gentamicin), the difference in potency was found to be very significant, with gentamicin being more effective with lower MIC and MBC values ranging from 0.33 to 2 µg/mL and 32 to 64 µg/mL, respectively.

### 2.5. Antifungal Activity

Methanolic extracts of four *C. mixtus* organs were evaluated for their antifungal properties against three human pathogens, including one yeast (*Candida albicans*) and two filamentous fungi (*Trichophyton rubrum* and *Aspergillus fumigatus*). The MIC and MFC values were determined and summarized in Table 11. The three fungal pathogens were tested at concentrations ranging from 1.25 to 40 mg/mL. The results showed that *C. mixtus* extracts appeared to be less effective against *C. albicans* and *A. fumigatus* than against *T. rubrum*. The fungus *A. fumigatus* was noted to be the most resistant to all extracts tested, with MIC and MFC values higher than 40 mg/mL. In the activity against *C. albicans*, the best result was reported by the stem extracts, with a MIC value of 40 mg/mL and MFC higher than 40 mg/mL, unlike the other extracts, which showed MIC and MFC values higher than 40 mg/mL. Concerning the activity against the *T. rubrum* strain, all extracts showed important antifungal activity, with the highest efficacy reported by the flower and leaf extracts (MIC = 1.25 and MFC = 2.5 mg/mL). Roots and stems extracts also showed antifungal properties, with MIC = 2.5 mg/mL and MFC = 5 mg/mL for roots and MIC = 3.57 mg/mL and MFC = 6.67 mg/mL for stems.

Compared to the reference antifungal voriconazole, the difference in activity was very significant; the latter was much more potent than the extract, with MIC and MFC values between 0.12 and 0.25 µg/mL and 0.66 and 4 µg/mL, respectively.

## 3. Discussion

Plant extracts containing phytochemicals are increasingly marketed as products with a positive impact on the human health system. This work aimed to characterize the metabolites of four organs from *C. mixtus* and evaluate their antioxidant, antibacterial, and antifungal activities.

Methanolic extracts of *C. mixtus* flowers showed the highest content of polyphenols (36%), followed by flavonoids (42%) (Figure 6). Polyphenol contents were similar in roots and stems (22%), followed by leaves (20%). For flavonoids, the stems showed a content close to that of the leaves (23% and 21%, respectively), followed by the roots (14%).

The total phenolic content of the *C. mixtus* flower extract found in this work (30.55 mg GAE/g DW) is slightly higher than that found in the flower extract of a similar study with Italian *Matricaria chamomilla* (2689.2 mg GAE/100 g DW) [18] and much higher than a study with Egyptian *M. chamomilla* (3.7 mg GAE/g DW) [19] and a commercial *M. chamomilla* product from the United Arab Emirates (21.4 mg GAE/g DW) [20].

In a previous study, aerial part extracts of *C. mixtus* (obtained from Bouznika, Morocco) presented a total phenolic content of 19.5 mg GAE/g DW in a methanolic extract and 38.2 mg GAE/g DW in an aqueous extract [21], showing a lower content than that found in current research (65.75 mg GAE/g DW). Elouaddari et al. [21] also reported that in aerial part extracts of *C. mixtus*, the total flavonoid content was 2.7 and 3.2 mg QE/g DW in aqueous and methanolic extracts, respectively. In this investigation, our estimation showed a much greater amount of flavonoid content (53.61 mg QE/g DW) (Table 1) than in the aforementioned study by Elouaddari et al. [21]. The differences in extraction methods and the solvent and timing used may influence the composition of polyphenols, flavonoids, and other compounds and therefore also affect their biological activities.

In the present study, we applied the endpoint method developed by Arnao et al. [22]. This technique uses 2,2′-azino-bis-3-ethylbenzthiazoline-6-sulfonic acid (ABTS) as a chromogen to estimate total antioxidant activity. It is a robust method, that is widely used and applied to various biological samples [23]. Although this method was originally developed for the study of plant foodstuffs, it can also be used to characterize plant extracts [16]. The ABTS^·+^ chromogen used in our method was compared to another widely used radical chromogen, which is 2,2-diphenyl-1-picrylhydrazyl (DPPH^·^). Furthermore, the extraction method and the solvent used can influence the composition of polyphenols, flavonoids, and other compounds and therefore also affect the antioxidant activity [19,24].

Zeroual et al. [25] showed that methanolic extracts of *C. mixtus* flowers had the highest free radical scavenging activity (IC_50_ = 55.50 µg/mL), followed by the ethanolic extract (IC_50_ = 121.5 µg/mL), ethyl acetate (IC_50_ = 240.9 µg/mL), and n-hexane (IC_50_ = 259 µg/mL). Similarly, the DPPH test showed that methanolic extracts of chamomile flowers (*Matricaria chamomilla*) had the strongest antiradical power (IC_50_ = 0.0022 µMoles) compared to ethanolic, diethyl ether, and hexane extracts [19]. In contrast, methanolic extracts of the aerial part of *C. mixtus* showed lower antioxidant activity than the aqueous extract by DPPH and ABTS tests [21].

Furthermore, the choice of extraction organ plays an important role in antioxidant activity. In our study, we chose to separate the plant organs (flowers, leaves, stems, and roots) to compare their antioxidant activity. Methanolic extracts of the roots showed the best activity with ABTS, while the flowers showed a better result with DPPH. On the other hand, the leaves of *C. mixtus* showed the lowest activity in both tests.

A correlation analysis was carried out to study the relationship between the phenolic and flavonoid contents of the extracts and their antioxidant activities (Table 12). Methanolic extracts of stem, leaf, and flower showed a very significant positive correlation between the content of phenolics and the antioxidant activity by the ABTS (r² = 0.93, *p* < 0.05) and DPPH (r^2^ = 0.94, *p* < 0.05) tests. These organs also showed a strong correlation between their total flavonoid content and antioxidant activity by the ABTS (r² = 0.89, *p* < 0.05) and DPPH (r^2^ = 0.90, *p* < 0.05) tests.

The ability of flavonoids to act as in vitro antioxidants has been the subject of several studies in recent years, and important structure-activity relationships for antioxidant activity have been established [26,27]. Almost all flavonoid groups can act as antioxidants, but it has been reported that flavones and catechins appear to be the most potent flavonoids in protecting the body against reactive oxygen species [28].

Moreover, many studies have reported the benefits of phenolic compounds, such as vanillin, that have antioxidant and antidepressant activities and neuroprotective, antimutagenic, and anticarcinogenic effects [29].

Based on the GC-MS analysis, the present research reveals that derivatized methanolic extracts from *C. mixtus* contain carbohydrates, lactones, fatty acids, organic acids, amino acids, terpenoids, alcohols, phenolics, alkanes, and other compounds, with carbohydrate compositions in abundance. Comparing these results with others obtained of methanolic extracts from the same plant without derivatization [11], we found that the flower extracts were dominated by fatty acids (27.86%), leaf extracts by terpenoids (46.20%), stem extracts by esters (30.11%), and root extracts by alcohols (24.49%) and esters (21.91%). The choice of the derivatization technique was made to improve and increase the volatility, sensitivity, thermal stability, greater selectivity, and separation behavior of the analytes [30]. Another study on aqueous extracts of two Moroccan chamomiles, *C. mixtus* and *M. chamomilla*, discovered the presence of alkaloids, terpenoids, saponins, flavonoids, and tannins, but not anthraquinones [31].

Elouaddari et al. (2019) [32] examined the chemical composition and biological activities of *C. mixtus* essential oils (EOs). According to the authors review, a total of 264 compounds constitute the EOs of *C. mixtus*, which vary greatly depending on diverse parameters, including geographies, plant parts, extraction methods, and ecological factors. The distribution of these chemicals is as follows: oxygenated monoterpenes (30–45.3%), sesquiterpene hydrocarbons (14–33.9%), monoterpene hydrocarbons (15–24.5%), sesquiterpenes (4–11.7%), and others (traces–4.5%). Many properties, including antimicrobial, anticorrosive, and cytotoxic activity against human cervical cancer cell lines, are caused by these biomolecules.

Our results of HPLC-MS analysis reported in the previous study [11] showed that the different methanolic extracts of the *C. mixtus* plant are very rich in glycosides and aglycones (luteolin, apigenin, luteolin-7-*O*-glucoside, apigenin-7-glucoside, quercetin, rutin, naringin, catechin, vanillin, kaempferol, and isorhamnetin) and phenolic acids (gallic, protocatechuic, chlorogenic, salicylic, p-hydroxybenzoic, caffeic, vanillic, syringic, methyl paraben, rosmarinic, p-coumaric, and ferulic acids). The presence of phenolic compounds and flavonoids may contribute to the activity of the extracts [33]. Flavonoids, such as epicatechin and rutin, have been reported to be powerful radical scavengers [28]. The scavenging ability of rutin may be due to its inhibitory activity on the xanthine oxidase enzyme [28]. We revealed the presence of rutin in all extracts of *C. mixtus* with interesting values, and the highest value was detected in *C. mixtus* flowers (673.12 µg/g DW) [11].

Additionally, the antimicrobial activities of natural products have attracted much attention due to the increasing incidence of pathogens that have become drug-resistant [34]. In this study, the antibacterial activity of methanolic extracts of the flowers, leaves, stems, and roots of *C. mixtus* was tested against *S. aureus* and *E. coli* strains isolated from clinical samples. *C. mixtus* extracts exhibited antibacterial activity against the two selected strains, with the best result obtained with the leaf extract. The other organs showed almost the same activity against *E. coli.*

However, extracts are more effective against Gram-positive bacteria than Gram-negative bacteria. This difference in sensitivity is due to membrane permeability. Gram-negative bacteria have a complex and rigid membrane rich in lipopolysaccharide, which can limit the passage of antimicrobial constituents [35].

The antibacterial activity of *Matricaria chamomilla* extract (Asteraceae family) from Djibouti was investigated [36]. Methanolic extracts of *M. chamomilla* leaves showed higher antibacterial activity against *E. coli* (MIC and MBC = 25 µg/mL) compared to *S. aureus* (MIC and MBC = 100 µg/mL).

Concerning the antifungal activity, against *C. albicans* and *A. fumigatus*, *C. mixtus* extracts appeared to be less effective, and *A. fumigatus* was the most resistant (MIC and MFC > 40 mg/mL). In another report, the antifungal activity of the aerial part (leaves and flowers) of *M. chamomilla* extracts (aqueous, methanol, and chloroform) was studied against *C. albicans* and *Fusarium* spp., and the results showed that the extracts had no effect on the fungal strains tested [37]. In reverse, methanolic extracts of *Montanoa* sp. and *Schistocarpha sinforosi* Cuatrec. from the Asteraceae family showed moderate activity against *C. albicans* (MICs = 0.62 and 2.50 mg/mL, respectively) [38].

In a study reported by Mekonnen et al. [39], the essential oil from *M. chamomilla* flowers had no inhibitory effect against all strains of *Trichophyton* and *Aspergillus*. Comparing these results with the present study, all the methanolic extracts of *C. mixtus* gave very good results against *T. rubrum*.

Accordingly, it can be inferred from our results that the extracts of both plants showed antibacterial and antifungal activity. This can be explained by their chemical composition, which is rich in phenolic acids and flavonoids. They are also very rich in terpenoids, fatty acids, organic acids, esters, and ketones [11].

Protocatechuic acid was isolated from the aerial parts of *Centaurea spruneri* of the Asteraceae family to test its antibacterial activity [40]. The MIC and MBC values against *S. aureus*, *E. coli*, *Bacillus cereus*, *Micrococcus flavus*, *Listeria monocytogenes*, *Pseudomonas aeruginosa*, *Proteus mirabilis*, and *Salmonella typhimurium* were between 100 and 400 mg/L. Several studies have revealed that other phenolic compounds have antimicrobial activities, such as ellagic acid [41], gallic acid [42,43], and p-hydroxybenzoic acid [44].

Overall, the findings of the present work demonstrated that all extracts of *C. mixtus* organs exhibited good antioxidant and antimicrobial activities, and results differed from one organ to another. Therefore, the extracts of the plant materials studied could be recommended as a source of pharmaceutical materials necessary for the preparation of new antioxidant and antimicrobial agents.

## 4. Materials and Methods

### 4.1. Standards and Chemical Reagents

Dimethyl sulfoxide (DMSO) was obtained from Sigma (Darmstadt, Germany). Mueller–Hinton Broth (MHB) and Mueller–Hinton Agar (MHA) media were purchased from Liofilchem (Teramo, Italy). RPMI-1640 broth medium was obtained from Biochrom AG (Berlin, Germany). Sabouraud dextrose agar (SDA) was purchased from BioMérieux (Marcy L’Étoile, France). 2,2′-Azino-bis-(3-ethylbenzthiazoline-6-sulfonic acid) (ABTS), 6-hydroxy-2,5,7,8-tetramethylchroman-2-carboxylic acid (trolox), horseradish peroxidase (HRP) type VI, and 2,2-diphenyl-1-picrylhydrazyl 95% (DPPH) were obtained from Sigma Chem. Co. (Madrid, Spain). H_2_O_2_ (30% *v*/*v*) was purchased from Aldrich Chem. Co. (Madrid, Spain). Methanol, N-O-bis(trimethylsilyl) trifluoroacetamide (BSTFA), and pyridine were obtained from Merck KGaA (Darmstadt, Germany).

### 4.2. Plant Material and Preparation of Methanolic Extracts

The *C. mixtus* plants were collected in May 2018 at full maturity from the Beni Hassane region, province of Tanger-Tetouan-Al Hoceima in northern Morocco (N 35°21′20.865″, W 5°22′12.677″) and brought to the laboratory. The identification of the plant was carried out by Prof. Ahmed Lamarti from the Faculty of Science in Tetouan (Morocco). The organs (roots, stems, leaves, and flowers) of the fresh plant were separated before the material was dried in an oven until it reached a stable dry weight at 50 °C. After that, it was ground at 8000 rpm in a Microtron MB 550 (Kinematica AG, Eschbach, Germany). The powder was made up of particles with a diameter of about 0.2 mm and was kept at room temperature in the dark.

Methanolic extraction was performed following the previous work by Barros et al. [45] with a slight modification. Two grams of fine dried powder from each plant part were extracted by stirring with 100 mL of methanol at 25 °C at 150 rpm for 24 h and were filtered afterward through Whatman N^o^ 4 paper. The filtration residue was extracted twice more using the same method. Methanolic extracts of each plant part were combined and dried using a rotary evaporator under vacuum (Rotavapor^®^ R-210, BÜCHI, Flawil, Switzerland) at 45 °C. Then, the dried extracts of roots, stems, leaves, and flowers were weighed and stored at −80 °C for further use. For each organ, the extraction yield was calculated.

### 4.3. Estimation of the Total Phenolic Content from Cladathus mixtus

The Folin–Ciocalteu reagent was used with some modifications to determine the total phenolic content (TPC) in each organ as described by Singleton and Rossi [46]. In a glass test tube, 50 µL of the sample was placed, followed by 950 µL of distilled water, 50 µL of 1 M sodium carbonate, and 50 µL of Folin–Ciocalteu reagent. After 15 min of standing in a water bath at 30 °C, the absorbance at 715 nm was measured. The results were given in milligrams of gallic acid equivalents per gram of dry weight (mg GAE g^−1^ DW). A Perkin-Elmer Lambda-2S UV-VIS spectrophotometer (Loughborough, UK) was used to take photometric measurements. Experiments were carried out in triplicate.

### 4.4. Estimation of the Total Flavonoids Content from Cladathus mixtus

The total flavonoid content (TFC) was determined by using an aluminum chloride colorimetric method modified by Woisky and Salatino [47] with a slight modification. Briefly, 0.5 mL of methanolic extract was mixed with 1.5 mL of 95% ethanol, 0.1 mL of 10% aluminum chloride, 0.1 mL of 1 M potassium acetate, and 2.8 mL of distilled water. The absorbance of the reaction mixture was measured at 415 nm after 30 min of incubation at room temperature. The findings were reported in mg of quercetin equivalents per gram of dry weight (mg QE/g DW). Moreover, the experiments were carried out in triplicate.

### 4.5. GC-MS Analysis of Methanolic Extracts from Cladathus mixtus

A chemical analysis of methanolic extracts from *C. mixtus* was performed using gas chromatography (GC) in a Trace 1300 gas chromatograph (Thermo Fisher Scientific, Waltham, MA, USA) coupled to mass spectrometry (MS) (ISQ single quadrupole mass spectrometer; Thermo Fisher Scientific) and an automatic injector. The GC was outfitted with a capillary column DB-5 (30 µm, 0.25 mm i.d., film thickness 0.25 µm) with a non-polar stationary phase (5% phenyl, 95% dimethylpolysiloxane) from Thermo Fisher Scientific (Waltham, MA, USA). The temperature of the column was programmed to rise from 50 to 350 °C at a rate of 5 °C/min. At a flow rate of 0.75 mL/min, helium was used as the carrier gas [48]. To perform GC-MS analysis, the dried methanolic extracts were derivatized by mixing 10 mg of each sample with 100 µL of anhydrous pyridine and 100 µL BSTFA, then the mixture was heated at 65 °C for 30 min and diluted with 200 µL chloroform. Finally, the derivatized solution (50 µL) was analyzed using GC-MS.

### 4.6. Estimation of Antioxidant Activity of the Cladanthus mixtus Extracts

#### 4.6.1. Free-Radical Scavenging Activity on 2,2-Diphenyl-1-Picrylhydrazyl (DPPH)

The DPPH* test is the oldest indirect method for determining antioxidant activity and was first used to determine the antioxidant potential of phenolic compounds [49]. DPPH* is a very stable radical chromogen that is acquired directly without preparation (ready to dissolve). It has a dark blue color and is a long-lived nitrogen radical species due to its inability to undergo dimerization [50]. A 0.1 mM DPPH solution was prepared in methanol. Of the DPPH stock solution, 900 μL were mixed with 100 μL of plant extract solution. Trolox was used as a reference standard. The reaction was performed in triplicate and allowed to stand at room temperature for 30 min. The decrease in absorbance at 517 nm, which is proportional to soaked (DPPH*), was determined in mg trolox equivalents per g dry weight (mg TE/g DW).

#### 4.6.2. 2,2′-Azino-Bis (3-Ethylbenzothiazoline-6-Sulfonic Acid (ABTS)

The antioxidant activity was measured using the ABTS/H_2_O_2_/HRP decoloration methods according to the procedure employed by Arnao et al. [22] and previously described by El Mihyaoui et al. [16]. The reaction mixture contained 1 mM ABTS, 75 µM hydrogen peroxide (H_2_O_2_), and 6 µM horseradish peroxidase (HRP) type VI in acidified ethanol (pure ethanol with phosphoric acid, 0.7% *w*/*v*), in a total volume of 1 mL prepared at 25 °C. A volume of 40 µL of each methanolic extract was added to the reaction medium, and the decrease in absorbance at 730 nm was measured after 6 min. The absorbance decrease was measured from the difference between the absorbance at 730 nm values before 6 min and after sample addition. Antioxidant activity was calculated as moles of ABTS^·+^ quenched by 1 mole of trolox. The results were expressed as trolox equivalents per gram of dry weight (mg TE/g DW). Experiments were conducted in triplicate.

### 4.7. Estimation of Antimicrobial Activities of the Cladanthus mixtus Extracts

#### 4.7.1. Microorganism Strains

Two strains of Gram-negative and Gram-positive bacteria were used to evaluate the antibacterial activity: *Staphylococcus aureus* (ATCC 25923) and *Escherichia coli* (ATCC 25922). Stock cultures were maintained on Mueller–Hinton broth (MHB) medium with 15% glycerol at −80 °C and subcultured on Mueller–Hinton agar (MHA) before each test.

The antifungal activity of methanolic extracts of *C. mixtus* was checked against three fungi: *Candida albicans* (ATCC 10231), *Aspergillus fumigatus* (ATCC 46645), and *Trichophyton rubrum* (clinical strain, FF 5). All microorganisms were stored in SDB (Sabouraud dextrose broth) with 20% glycerol at −80 °C and subcultured in SDA (Sabouraud dextrose agar) or potato dextrose agar (PDA) before each test to provide pure and ideal growing conditions. The strains were obtained from the Department of Microbiology, Faculty of Pharmacy, Porto University (Portugal), where all the susceptibility tests were completed.

#### 4.7.2. Broth Microdilution Method

Broth microdilution method, based on the Clinical and Laboratory Standards Institute (CLSI) (M07-A8, bacteria; M27-A3, yeasts; and M38-A2, filamentous fungi) and previously described by Erbiai et al. [51], was used to determine the antimicrobial activities of *C. mixtus* extracts. In brief, the extracts were dissolved in 25% dimethyl sulfoxide (DMSO), and serial dilutions were prepared in MHB for bacteria and in RPMI-1640 for fungi. Of each concentration medium, 100 µL was then distributed into sterile 96-well plates, followed by 100 µL of the final cell suspension (1–2 × 10^5^ CFU/mL for bacteria, 1–5 × 10^3^ CFU/mL for yeasts, 0.4–5 × 10^4^ CFU/mL for Aspergillus, and 1–3 × 103 CFU/mL for dermatophytes), which was diluted in fresh MHB for bacteria and RPMI 1640 for fungi. Then, the plates were incubated without agitation at 37 °C for 24 h for bacteria, 48 h for *C. albicans* and *A. fumigatus*, and for seven days at 25 °C for *T. rubrum*. The same organisms were also tested against the reference antibacterial drug, gentamicin, and the reference antifungal drug, voriconazole, for comparison of results and quality control.

The concentration that induced no visible growth was referred to as the minimal inhibitory concentration (MIC). To estimate the minimum bactericidal (MBC) and fungicidal concentration*s* (MFC), 10 µL from the wells with no turbidity were inoculated into a Petri dish containing MHA medium for bacteria and SDA medium for fungi. Under the previously specified incubation conditions, the MBC and MFC were determined as the lowest concentrations that completely inhibited the development of the tested strains.

### 4.8. Statistical Analysis

The SPSS program (Chicago, IL, USA) was used, applying a one-way ANOVA to evaluate the statistical differences among the group and the Tukey multiple range test to establish significant differences between the evaluated parameters. The results of three independent experiments are represented as the mean ± SD for extraction yield, phenolic and flavonoid contents, and antioxidant and antimicrobial activities.

## 5. Conclusions

This is the first comparative study on the four organs of the *C. mixtus* plant from the Beni Hassane region, province of Tanger-Tetouan-Al Hoceima, in northern Morocco. The total content of phenolic and flavonoid compounds from methanolic extracts was highly significant. Therefore, it can be concluded that *C. mixtus* is rich in phenolics and flavonoids, mainly in the flowers. Furthermore, many biomolecules were identified in derivatized methanolic extracts using GC-MS analysis. In addition, the extracts showed strong antioxidant activity, which was correlated with their phenolic and flavonoid contents. The extracts also showed antimicrobial activity, with *S. aureus* being more sensitive than *E. coli*. Moreover, the extracts were less effective against *C. albicans* and *A. fumigatus* than against *T. rubrum*. All extracts showed good antifungal properties against the *T. rubrum* strain, whereas flower and leaf extracts were the most effective. Overall, the important biological properties of *C. mixtus* were due to its richness in many bioactive compounds, which differed from one organ to another. Herein, this research suggests that *C. mixtus* exhibits interesting health-related bioactivities, but more specific research should be performed to determine the phytotherapeutic and dietary applications of interest.

## Figures and Tables

**Figure 1 molecules-28-03196-f001:**
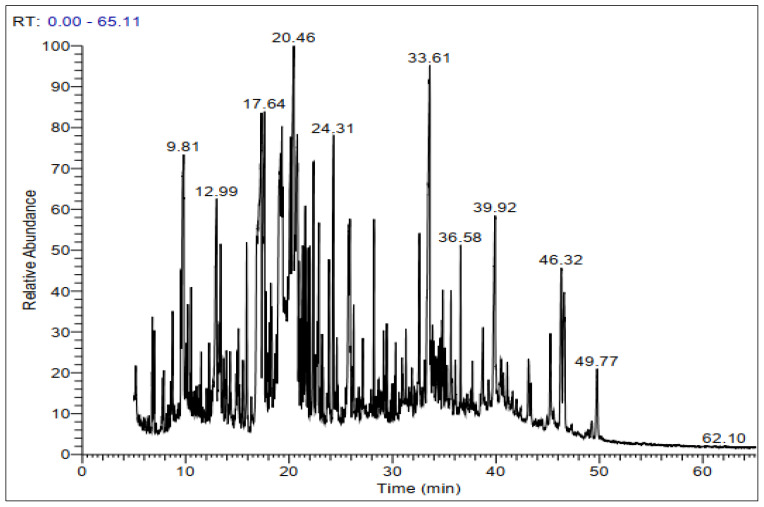
Representative GC-MS chromatogram of derivatized methanolic extract of *C. mixtus* flowers.

**Figure 2 molecules-28-03196-f002:**
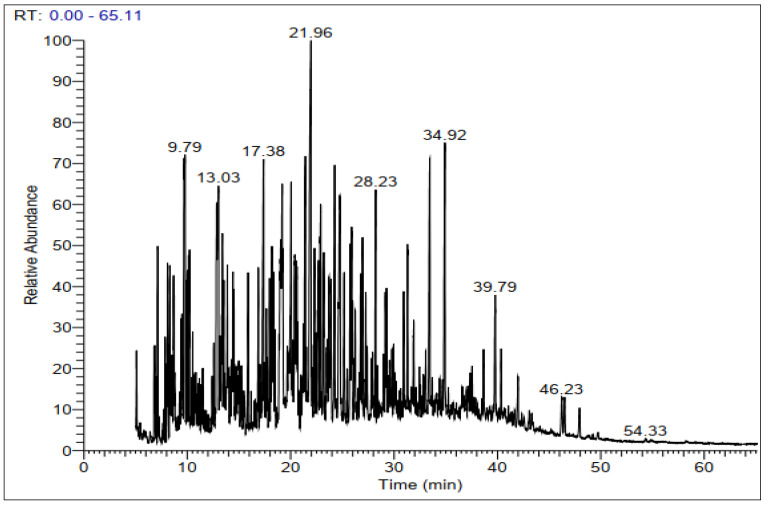
Representative GC-MS chromatogram of derivatized methanolic extract of *C. mixtus* leaves.

**Figure 3 molecules-28-03196-f003:**
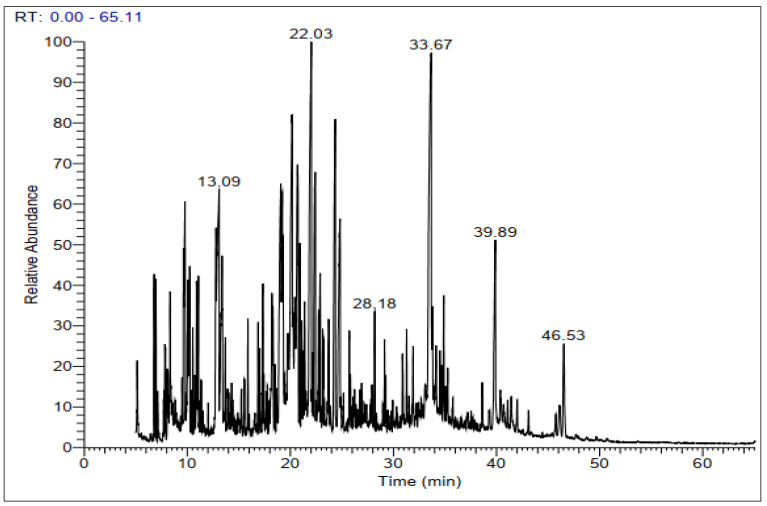
Representative GC-MS chromatogram of derivatized methanolic extract of *C. mixtus* stems.

**Figure 4 molecules-28-03196-f004:**
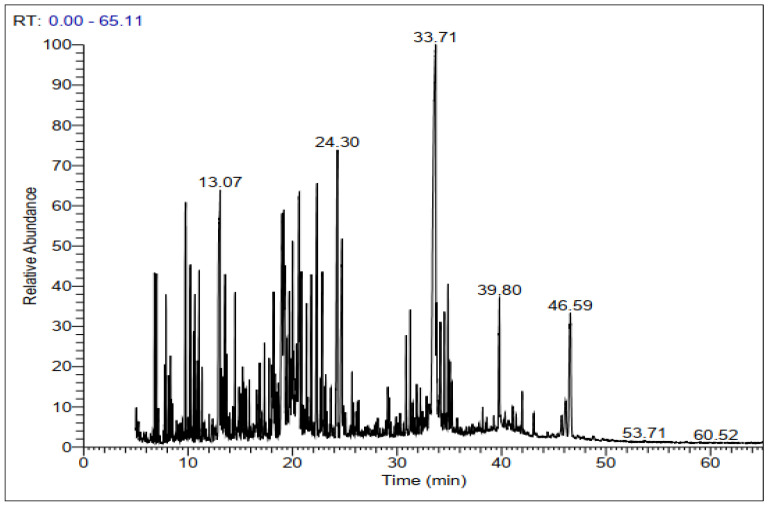
Representative GC-MS chromatogram of derivatized methanolic extract of *C. mixtus* roots.

**Figure 5 molecules-28-03196-f005:**
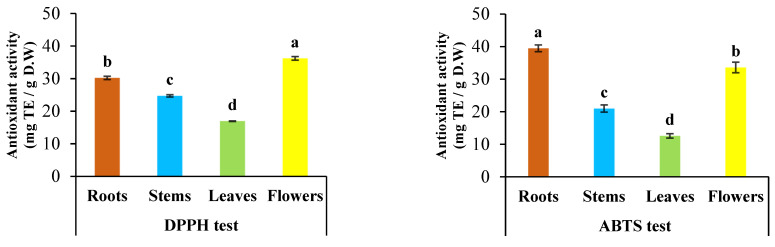
Antioxidant activity of different organ extracts of *Cladanthus mixtus* by the DPPH and ABTS tests. Different letters indicate significant differences between the organs of one plant at *p* < 0.05. Values are means ± S.D. for three replicates.

**Figure 6 molecules-28-03196-f006:**
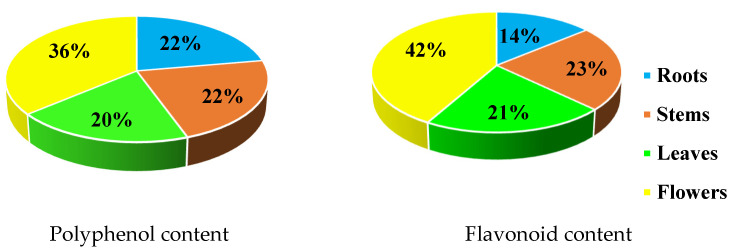
Percentage of polyphenols and flavonoids in methanolic extracts of *Cladanthus mixtus*.

**Table 1 molecules-28-03196-t001:** Extraction yield, polyphenol, and flavonoid contents in methanolic extracts of *Cladanthus mixtus*.

Organs	Extraction Yield (%)	Polyphenols (mg GAE/g DW)	Flavonoids (mg QE/g DW)
Roots	20.65 ± 2.86	18.83 ± 1.04 ^b,^*	8.74 ± 0.33 ^c^
Stems	18.75 ± 3.41	18.77 ± 0.39 ^b^	14.37 ± 0.55 ^b^
Leaves	19.40 ± 4.11	16.43 ± 0.32 ^b^	13.24 ± 0.05 ^b^
Flowers	25.86 ± 0.73	30.55 ± 0.85 ^a^	26.00 ± 1.34 ^a^

* Values are expressed as the mean ± SD of three replicates. Different letters indicate significant differences between the organs in the same column at *p* < 0.05.

**Table 2 molecules-28-03196-t002:** Biomolecule groups of derivatized methanolic extracts from *Cladanthus mixtus* analyzed by GC-MS.

	Area (%)			
Compound Groups	Flowers	Leaves	Stems	Roots
**Carbohydrates**	27.25	54.80	64.87	62.57
**Lactones**	24.60	--	--	--
**Organic acids**	11.81	18.82	18.68	13.00
**Fatty acids**	8.13	9.08	1.58	3.18
**Phenols**	7.08	1.50	8.46	6.66
**Amino acids**	--	7.10	1.26	2.28
**Others**	21.13	8.69	5.15	12.31
Total	100	99.99	100	100

(--): Not detected.

**Table 3 molecules-28-03196-t003:** Carbohydrate composition of derivatized methanolic extracts obtained by GC-MS.

			Area (%)			
Molecules	Chemical Formula	Molecular Weight	Flowers	Leaves	Stems	Roots
Sucrose	C_12_H_22_O_11_	342.2	17.01	0.13	14.84	22.47
Myo-Inositol	C_6_H_12_O_6_	180.1	--	9.38	13.9	6.65
Xylitol	C_5_H_12_O_5_	152.1	--	6.15	0.66	--
Trehalose	C_12_H_22_O_11_	342.3	1.22	4.61	0.98	3.93
Glucose	C_6_H_12_O_6_	180.1	--	6.39	0.25	0.23
D-Fructofuranose	C_6_H_12_O_6_	180.1	--	3.19	7.37	8.46
α-Lactose	C_12_H_22_O_11_	342.3	0.63	2.04	0.38	0.23
D-Allofuranose	C_6_H_12_O_6_	180.1	--	--	0.03	5.22
Glyceric acid	C_3_H_6_O_4_	106.0	0.84	0.41	0.59	0.59
α-Mannobiose	C_12_H_22_O_11_	342.1	2.00	0.33	0.79	0.21
D-Galacturonic acid	C_6_H_10_O_7_	194.1	0.74	2.14	0.52	--
D-Ribofuranose	C_5_H_10_O_5_	150.1	1.40	--	0.20	1.65
Lactitol	C_12_H_24_O_11_	344.3	1.74	--	--	--
Turanose	C_12_H_22_O_11_	342.3	1.57	0.51	1.78	0.73
Palatinose	C_12_H_22_O_11_	342.3	0.10	--	--	--
D-Glucosamine	C_6_H_13_NO_5_	179.1	--	0.69	--	--
meso-Erythritol	C_4_H_10_O_4_	122.1	--	6.60	--	--
Levoglucosan	C_6_H_10_O_5_	162.1	--	1.53	--	--
D-Glucitol	C_6_H_14_O_6_	182.1	--	3.08	4.47	--
D-Gluconic acid	C_6_H_12_O_7_	196.1	--	1.89	2.29	0.47
Valproic acid glucuronide	C_14_H_24_O_8_	320.3	--	1.34	--	--
D-Glucopyranose	C_6_H_12_O_6_	180.1	--	2.11	--	--
Galactinol	C_12_H_22_O_11_	342.3	--	1.65	--	0.16
α-D-Mannopyranose	C_6_H_12_O_6_	180.1	--	0.63	--	--
Glycerol	C_3_H_8_O_3_	92.0	--	--	3.14	--
Arabinonic acid, 1,4-lactone	C_5_H_8_O_5_	148.1	--	--	0.21	--
D-Glucuronic acid	C_6_H_10_O_7_	194.1	--	--		0.34
D-Glucuronic acid-γ-lactone	C_6_H_8_O_6_	176.1	--	--	0.61	--
3-Deoxy-ribo-hexonic acid, 1,4-lactone	C_6_H_10_O_5_	162.1	--	--	2.14	--
D-Glucopyranose	C_6_H_12_O_6_	180.1	--	--	3.75	3.79
D-Galactopyranoside (= D-galactose)	C_6_H_12_O_6_	180.1	--	--	0.56	--
Methyl galactoside	C_7_H_14_O_6_	194.1	--	--	0.33	--
1,5-Anhydrohexitol	C_6_H_12_O_5_	164.1	--	--	1.01	--
Glyceryl-glycoside	C_27_H_66_O_8_Si_6_	687.3	--	--	0.64	--
Arabinofuranose	C_5_H_10_O_5_	150.1	--	--	0.66	--
D-Arabinopyranose	C_5_H_10_O_5_	150.1	--	--	--	0.34
D-Allopyranose	C_6_H_12_O_6_	180.1	--	--	0.62	1.22
D-Tagatofuranose	C_6_H_12_O_6_	180.1	--	--	0.24	--
D-Arabitol	C_5_H_12_O_5_	152.1	--	--	1.91	1.10
D-Ribofuranose	C_5_H_10_O_5_	150.1	--	--	--	0.46
1,5-Anhydrohexitol	C_6_H_12_O_5_	164.1	--	--	--	0.92
D-Psicose	C_6_H_12_O_6_	180.1	--	--	--	1.11
Dulcitol	C_6_H_14_O_6_	182.1	--	--	--	0.84
α-Glucopyranose phosphate	C_6_H_13_O_9_P	260.1	--	--	--	0.22
Beta-D-Lactose, (isomer 2)	C_12_H_22_O_11_	342.3	--	--	--	0.11
D-Psicofuranose	C_6_H_12_O_6_	180.1	--	--	--	0.16
Maltose	C_12_H_22_O_11_	342.3	--	--	--	0.39
Dihydroxyacetone	C_3_H_6_O_3_	90.0	--	--	--	0.58
Total			27.25	54.8	64.87	62.57

(--): Not detected.

**Table 4 molecules-28-03196-t004:** Lactones of derivatized methanolic extracts obtained by GC-MS.

			Area (%)			
Molecules	Chemical Formula	Molecular Weight	Flowers	Leaves	Stems	Roots
D-Glucurono-γ-lactone (isomer 1)	C_6_H_8_O_6_	176.12	24.26	--	--	--
Erythrono-1,4-lactone	C_4_H_6_O_4_	118.09	0.34	--	--	--
Total			24.60	--	--	--

(--): Not detected.

**Table 5 molecules-28-03196-t005:** Organic acids of derivatized methanolic extracts obtained by GC-MS.

			Area (%)			
Molecules	Chemical Formula	Molecular Weight	Flowers	Leaves	Stems	Roots
Malic acid	C_4_H_6_O_5_	134.0	9.16	6.02	12.26	6.67
Malonic acid	C_3_H_4_O_4_	104.0	1.06	1.45	--	--
Lactic acid	C_3_H_6_O_3_	90.0	0.76	0.45	0.83	0.71
Methyl malonate	C_8_H_14_O_4_	174.1	0.43	--	--	--
Glycolic acid	C_2_H_4_O_3_	76.0	0.40	0.30	0.57	0.67
3-Hydroxypropionic acid	C_3_H_6_O_3_	90.0	--	0.38	0.61	0.6
Succinic Acid	C_4_H_6_O_4_	118.0	--	2.12	1.85	--
3-Hydroxyisobutyric acid	C_4_H_8_O_3_	104.1	--	1.46	0.52	0.38
3-Hydroxyadipic acid	C_6_H_10_O_5_	162.1	--	0.51	--	--
Citric acid	C_6_H_8_O_7_	192.1	--	3.71	--	--
Quininic acid	C_11_H_9_NO_3_	203.1	--	2.42	--	1.90
Phosphoric acid, monomethyl ester	CH_5_O_4_P	112.0	--	--	--	0.16
Propyl acetate	C_5_H_10_O_2_	102.1	--	--	0.93	--
Fumaric acid	C_4_H_4_O_4_	116.0	--	--	0.09	--
2-Furoic acid	C_5_H_4_O_3_	112.0	--	--	--	0.43
Methyl 3-methoxyacrylate	C_5_H_8_O_3_	116.1	--	--	--	0.44
Pyruvic acid	C_3_H_4_O_3_	88.06	--	--	--	0.64
(R)-3-Hydroxybutyric acid	C_4_H_8_O_3_	104.1	--	--	--	0.40
Total			11.81	18.82	17.66	13.00

(--): Not detected.

**Table 6 molecules-28-03196-t006:** Fatty acids of derivatized methanolic extracts obtained by GC-MS.

			Area (%)			
Molecules	Chemical Formula	Molecular Weight	Flowers	Leaves	Stems	Roots
Linoelaidic acid	C_18_H_32_O_2_	280.4	6.73	1.63	--	0.38
Palmitic Acid	C_16_H_32_O_2_	256.4	--	2.45	--	1.83
Stearic acid	C_18_H_36_O_2_	284.4	1.40	0.73	--	0.18
Dimethyl malate	C_6_H_10_O_5_	162.1	--	--	1.05	--
Elaidic Acid	C_18_H_34_O_2_	282.5	--		--	0.22
Hexanoic acid (Caproic acid)	C_6_H_12_O_2_	116.1	--		--	0.18
Docosahexaenoic acid	C_22_H_32_O_2_	328.5	--	2.71	--	--
β-D-Glucopyranosiduronic acid, 3-(5-ethylhexahydro-2,4,6-trioxo-5-pyrimidinyl)-1,1-dimethylpropyl 2,3,4-tris-O-(trimethylsilyl)-, methyl ester	C_27_H_52_N_2_O_10_Si_3_	649.0	--	0.16	--	--
2-Hydroxybut-2-enedioic acid	C_4_H_4_O_5_	132.0	--	0.56	--	--
1-Monopalmitin	C_19_H_38_O_4_	330.5	--	0.84	0.53	0.39
Total			8.13	9.08	1.58	3.18

(--): Not detected.

**Table 7 molecules-28-03196-t007:** Phenolics of derivatized methanolic extracts obtained by GC-MS.

			Area (%)			
Molecules	Chemical Formula	Molecular Weight	Flowers	Leaves	Stems	Roots
Chlorogenic acid	C_16_H_18_O_9_	354.3	5.92	1.50	4.03	2.26
Caffeic acid	C_9_H_8_O_4_	180.1	--	--	4.43	4.01
Naringenin	C_15_H_12_O_5_	272.2	1.16	--	--	--
Isovanillic acid	C_8_H_8_O_4_	168.1	--	--	--	0.39
Total			7.08	1.50	8.46	6.66

(--): Not detected.

**Table 8 molecules-28-03196-t008:** Amino acids of derivatized methanolic extracts obtained by GC-MS.

			Area (%)			
Molecules	Chemical Formula	Molecular Weight	Flowers	Leaves	Stems	Roots
L-Homoserine	C_4_H_9_NO_3_	119.1	--	1.28	--	--
Proline	C_5_H_9_NO_2_	115.1	--	1.52	--	--
Serine	C_3_H_7_NO_3_	105.0	--	1.37	--	--
Threonine	C_4_H_9_NO_3_	119.1	--	1.44	0.62	--
Leucine	C_6_H_13_NO_2_	131.1	--	0.34	--	--
Pidolic acid	C_5_H_7_NO_3_	129.1	--	1.15	--	1.91
DL-Homophenylalanine	C_10_H_13_NO_2_	179.2	--	--	--	0.37
Sarcosine	C_3_H_7_NO_2_	89.0	--	--	0.64	--
Total			--	7.10	1.26	2.28

(--): Not detected.

**Table 9 molecules-28-03196-t009:** Chemical composition of other molecules of derivatized methanolic extracts obtained by GC-MS.

			Area (%)			
Molecules	Chemical Formula	Molecular Weight	Flowers	Leaves	Stems	Roots
Ethaneperoxoic acid, 1-cyano-4,4-dimethyl-1-phenylpentyl ester	C_16_H_21_NO_3_	275.3	0.24	--	--	--
4-Methyl-2-(2-nitro-5-piperidin-1-yl-phenyl)-2H-phthalazin-1-one	C_20_H_20_N_4_O_3_	364.4	--	1.84	--	--
Salbutamol	C_13_H_21_NO_3_	239.3	--	--	0.05	--
2-(2-Methoxyphenyl)propan-2-ol	C_10_H_14_O_2_	166.2	--	--	--	0.13
Terephthalic acid, di(4,4-dimethylpent-3-yl) ester	C_22_H_34_O_4_	362.5	--	--	--	1.98
N-Isopropyl-2-[4-(1,2,3-thiadiazol-4-yl) phenyl]hydrazine-1-carboxamide	C_12_H_15_N_5_OS	277.3	--	--	--	1.31
Propanedinitrile, 2-(5-phenylthio-2-thienylmethylene)-	C_14_H_8_N_2_S_2_	268.4	--	--	--	0.48
Phosphonic acid	H_2_O_3_P+	80.9	--	--	1.02	--
Phytol	C_20_H_40_O	296.5	--	1.24	--	
Stigmasterol	C_29_H_48_O	412.7	--	0.53	0.30	0.43
β-Sitosterol	C_29_H_50_O	414.7	--	0.19	0.21	0.24
Aucubin	C_15_H_22_O_9_	346.3	--	0.54	--	--
Lupenyl acetate	C_32_H_52_O_2_	468.8	0.37	0.05	--	--
Ethyl cholate	C_26_H_44_O_5_	436.6	1.14	--	--	--
alpha-Amyrin	C_30_H_50_O	426.7	0.54	--	--	--
Lanost-8-en-26-oic acid, 3α,12α-dihydroxy-24-methylene-, methyl ester	C_34_H_54_O_5_	542.7		--	--	0.02
1,2,4,5-Cyclohexanetetrol	C_6_H_12_O_4_	148.1	0.66	--	--	--
5-p-Coumaroylquinic acid	C_16_H_18_O_8_	338.1	0.80	--	--	--
1,2-Ethenediol	C_2_H_4_O_2_	60.0	0.04	--	--	--
Tetracosane	C_24_H_50_	338.6	0.64	--	--	--
Octadecylcyclohexane	C_24_H_48_	336.6	0.04	--	--	--
5-Methyluridine	C_10_H_14_N_2_O_6_	258.2	3.68	0.63	--	--
5-Methylcytosine	C_5_H_7_N_3_O	125.1	0.53	--	--	0.81
Deanol	C_4_H_11_NO	89.1	1.17	--	0.70	--
Copper phthalocyanine	C_32_H_16_CuN_8_	576.1	1.01	--	--	--
Silanol, trimethyl-, phosphate (3:1)	H_3_PO_4_ or H_3_O_4_P	97.995	10.27	--	--	3.20
17-Pentatriacontene	C_35_H_70_	490.9	--	0.06	--	--
Tricyclo [20.8.0.0(7,16)]triacontane, 1(22),7(16)-diepoxy-	C_30_H_52_O_2_	444.7	--	0.06	--	--
6-Ethoxy-4-methylquinoline-2-thiol	C_12_H_13_NOS	219.3	--	1.50	--	--
Uridine	C_9_H_12_N_2_O_6_	244.2	--	0.34	--	0.15
2-Oxa-3-azabicyclo [4.4.0]dec-3-ene, 5-methyl-1-trimethylsilyloxy-, N-oxide	C_12_H_23_NO_3_Si	257.0	--	0.69	--	--
2-Methoxyethyl(trimethyl)silane	C_6_H_16_OSi	132.2	--	0.62	--	--
Emulphor	C_20_H_40_O_2_	312.5	--	0.40		
1,3-Propanediol	C_3_H_8_O_2_	76.0	--	--	0.95	--
Ethylene glycol	C_2_H_6_O_2_	62.0	--	--	0.32	--
Dodecamethyl-pentasiloxane	C_12_H_36_O_4_Si_5_	384.8	--	--	1.60	--
2,3-Dihydroxypropyl acetate	C_5_H_10_O_4_	134.1	--	--	0.20	--
Glycerol monostearate	C_21_H_42_O_4_	358.6	--	--	0.27	--
Cortisol	C_21_H_30_O_5_	362.5	--	--	0.53	--
Hexadecyl (E)-m-coumarate					0.02	0.03
Isopropyl alcohol	C_3_H_8_O	60.1	--	--	--	0.23
1-Octanol	C_8_H_18_O	130.2	--	--	--	0.14
2-Butene-1,4-diol	C_4_H_8_O_2_	88.1	--	--	--	1.09
2-(2-Chloroethyl)-1-methylpyrrolidine	C_7_H_14_ClN	147.6	--	--	--	0.23
Acetamide	C_2_H_5_NO	59.0	--	--	--	0.06
Silane, diethylheptyloxyisobutoxy-	C_15_H_34_O_2_Si	274.5	--	--	--	1.00
α-Glycerophosphoric acid	C_3_H_9_O_6_P	172.0	--	--	--	0.78
Total			21.13	8.69	5.15	12.31

(--): Not detected.

**Table 10 molecules-28-03196-t010:** MIC and MBC values of methanolic extracts of *Cladanthus mixtus* and a reference antibiotic agent.

Methanolic Extracts (mg/mL)	Bacterial Species			
	Gram-Positive *Staphylococcus aureus*		Gram-Negative *Escherichia coli*	
	MIC	MBC	MIC	MBC
Roots	40.00 ± 0.00	40.00 ± 0.00	40.00 ± 0.00	40.00 ± 0.00
Stems	40.00 ± 0.00	40.00 ± 0.00	40.00 ± 0.00	>40.00
Leaves	20.00 ± 0.00	20.00 ± 0.00	30.00 ± 10.00	35.00 ± 8.66
Flowers	32.00 ± 9.80	32.00 ± 9.80	40.00 ± 0.00	40.00 ± 0.00
Gentamicin (µg/mL)	0.33 ± 0.11	32.00 ± 1.63	2.00 ± 0.00	64.00 ± 3.26

The results are presented as mean ± SD (*n* = 6).

**Table 11 molecules-28-03196-t011:** MIC and MFC values of methanolic extracts of *Cladanthus mixtus* and a reference antifungal agent.

Methanolic Extracts (mg/mL)	Fungal Species					
	*Candida albicans*		*Trichophyton rubrum*		*Aspergillus fumigatus*	
	MIC	MFC	MIC	MFC	MIC	MFC
Roots	>40	>40	2.5 ± 0.00	5.00 ± 0.00	>40	>40
Stems	40.00 ± 0.00	>40	3.57 ± 1.24	6.67 ± 2.36	>40	>40
Leaves	>40	>40	1.25 ± 0.00	2.5 ± 0.00	>40	>40
Flowers	>40	>40	1.25 ± 0.00	2.5 ± 0.00	>40	>40
Voriconazole (µg/mL)	0.25 ± 0.00	4.00 ± 0.00	0.12 ± 0.00	1.66 ± 0.47	0.25 ± 0.00	0.66 ± 0.23

The results are presented as the mean (*n* = 6).

**Table 12 molecules-28-03196-t012:** Correlation between antioxidant activities and total phenol and flavonoid contents of *Cladanthus mixtus* extracts.

Parameters Relationship	Organ	Equation	r^2^ *
		DPPH Test	
Polyphenol content vs. antioxidant activity	Stems, leaves, and flowers	y = 1.2486x − 1.5089	0.94
Flavonoid content vs. antioxidant activity	Stems, leaves, and flowers	y = 1.3098x + 2.4508	0.90
		ABTS Test	
Polyphenol content vs. antioxidant activity	Stems, leaves, and flowers	y = 1.3533x − 7.2704	0.93
Flavonoid content vs. antioxidant activity	Stems, leaves, and flowers	y = 1.4176x − 2.9432	0.89

* Indicates a significant correlation at *p* < 0.05.

## Data Availability

Data is contained within the article.

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
