# Peer review of "Chemical Characterization and Several Bioactivities of Cladanthus mixtus from Morocco"

_molecules, 2023, doi:10.3390/molecules28073196_

Round 1
Reviewer 1 Report
This manuscript described the Chemical characterization and several bioactivities of Cladanthus mixtus from Morocco. The major compounds were characterized by GC-MS and no new compounds were identified. The authors reported the antioxidant and antibacterial activities. But all the bioassays were done on mixtures and the activities were weak. So the manuscript can not be accepted.
Author Response
Comments and Suggestions for Authors.
This manuscript described the chemical characterization and several bioactivities of Cladanthus mixtus from Morocco. The major compounds were characterized by GC-MS and no new compounds were identified. The authors reported the antioxidant and antibacterial activities. But all the bioassays were done on mixtures and the activities were weak. So the manuscript can not be accepted.
R: We cannot agree with your final recommendation. Our MS describes globally and particularly all the compounds analyzed by the GC-MS, which complemented the data previously obtained by LC-MS techniques, and published in Cancers journal (https://doi.org/10.3390/cancers15010152), providing a detailed qualitative and quantitative chemical description of C. mixtus plants of Moroccan origin. Regarding the assessments of antioxidant and antibacterial activities, our data indicate relevant activities in both cases (see ref. 16 of reference list), discriminating between the four organs of the plant separately as a novelty; and using complete extracts and not individual chemical substances, on which there are many studies. Finally, we participate in the constructive purpose of the editor and other reviewers on the possibilities of improving our MS.
Reviewer 2 Report
The paper contains new and in-depth chemical and biological studies about Cladanthus mixtures from Morocco. The abstract is well written and organized. Additionally, the biological activities were investigated. However, Major revisions are required.
1- In abstract, please add more qualitative and quantitate values for the results part . the abstract contain no numerical values.
2- In the abstract, the authors should highlight the merits for this research over old published article for the same plant. Novelty statement is required.
3- The following articles should be mentioned and cited in the text
A. Wild Chamomile [Cladanthus mixtus (L.) Chevall.] Collected from Central-Northern Morocco: Phytochemical Profiling, Antioxidant, and Antimicrobial Activities; January 2021 Biointerface Research in Applied Chemistry 11(4):11440-11457 DOI: 10.33263/BRIAC114.1144011457
B. Chemical Composition and Biological Activities of the Cladanthus mixtus Essential Oil: A Review. September 2019, Analytical Chemistry Letters 9(5):649-663, DOI: 10.1080/22297928.2019.1682665
C. https://academicjournals.org/journal/JMPR/article-full-text/52061CE67892
D. CHEMICAL COMPOSITION AND ANTIMICROBIAL ACTIVITY OF THE ESSENTIAL OIL OF CHAMAEMELUM NOBILE (L.) ALL. https://pharmacologyonline.silae.it/files/archives/2021/vol2/PhOL_2021_2_A050_MohamedAbdoul-Latif.pdf
4- The aim of the work should be extended to compare the research outcomes with old related articles.
5- Merits of the new study over the aforementioned article should be stated in the introduction or the discussion
6- Full names for abbreviations stated in lines 89-96 are needed e.g. GAE, DW, QE. Additionally, full abbreviation list is required
Also for MIC and MBC
7- Correct the figure number in line 189;
Figure 2. Antioxidant activity of different organs extracts of Cladanthus mixtus by the DPPH and ABTS tests. Different letters 189 indicate significant differences between the organs for one plant at p < 0.05. Values are means ± S.D. for 3 replicates.
It should be figure 5.
8- Additionally, the authors should comment for the non-agreement among results of DPPH and ABTS tests?.
9- In lines 244-250; Give reasons for the differences in results obtained if compared to old reported method Elouaddari et al. [20],
10- Write full name for abbreviation AA in table 12.
11- In line 339, the present study, all the methanolic extracts of C. mixtus gave very good results against T. rubrum.
Did the authors tested methanol as a blank solvent to estimate that the activity is related to the plant and not the methanol itself?
This is very important especially old papers confirmed that the plant extract has no inhibitory activity as stated in ref 37.
12- Dervitization reagents should be stated in the abstract. This point is of interest for readers
Best wishes
Author Response
Comments and Suggestions for Authors.
The paper contains new and in-depth chemical and biological studies about Cladanthus mixtures from Morocco. The abstract is well written and organized. Additionally, the biological activities were investigated. However, Major revisions are required.
1- In abstract, please add more qualitative and quantitate values for the results part.the abstract contain no numerical values.
R: Thank you very much, we appreciate your suggestion, the quantitative information suggested was added, lines 34-47.
2-In the abstract, the authors should highlight the merits for this research over old published article for the same plant. Novelty statement is required.
R: Thank you for your comment. We highlighted the merits for this research in lines 25-27.
3-The following articles should be mentioned and cited in the text
A.Wild Chamomile [Cladanthus mixtus (L.) Chevall.] Collected from Central-Northern Morocco: Phytochemical Profiling, Antioxidant, and Antimicrobial Activities; January 2021 Biointerface Research in Applied Chemistry 11(4):11440-11457 DOI: 10.33263/BRIAC114.1144011457
B.Chemical Composition and Biological Activities of the Cladanthus mixtus Essential Oil: A Review. September 2019, Analytical Chemistry Letters 9(5):649-663, DOI: 10.1080/22297928.2019.1682665
- https://academicjournals.org/journal/JMPR/article-full-text/52061CE67892
D.CHEMICAL COMPOSITION AND ANTIMICROBIAL ACTIVITY OF THE ESSENTIAL OIL OF CHAMAEMELUM NOBILE (L.) ALL. https://pharmacologyonline.silae.it/files/archives/2021/vol2/PhOL_2021_2_A050_MohamedAbdoul-Latif.pdf
R: Thank you very much for your suggestions. We have added reference B (line 318) [32] and D (line 67) [9]. The articles A (line 276) [25] and C (line 76) [16] really were cited in the original version.
4-The aim of the work should be extended to compare the research outcomes with old, related articles.
R: Thank you very much for your comment. A few studies have been carried out on Cladanthus mixtus in general and particularly on specific organ extracts. For this raison, we compared our data with the results of the Matricaria chamomilla. Although it was part of another genus but both plants are very similar botanically, and both considered as chamomiles.
5-Merits of the new study over the aforementioned article should be stated in the introduction or the discussion.
R: Thank you very much for you comment. To show the merits of our study we have incorporated in the new version (lines 91-94) this paragraph:
To the best of our knowledge, this is the first comparative research study on the chemical characterization and biological activities of different C. mixtus organs (flowers, leaves, stems, and roots) from northern Morocco (Tangier-Tetouan-Al Hoceima region).
6-Full names for abbreviations stated in lines 89-96 are needed e.g. GAE, DW, QE. Additionally, full abbreviation list is required. Also for MIC and MBC.
R: Thank you for your comment. The information needed was added to all abbreviations mentioned first time. Also, an abbreviation list has been incorporated at the end of text.
7-Correct the figure number in line 189;
Figure 2. Antioxidant activity of different organs extracts of Cladanthus mixtus by the DPPH and ABTS tests. Different letters 189 indicate significant differences between the organs for one plant at p < 0.05. Values are means ± S.D. for 3 replicates.
It should be figure 5.
R: Thank you very much for you comment, we corrected this error.
8-Additionally, the authors should comment for the non-agreement among results of DPPH and ABTS tests?
R: ABTS and DPPH are antioxidant activity tests with different mechanistic approaches, although the main difference is their different response against to the different solvents used. Thus, while DPPH is applied using alcoholic solvents (not in water), ABTS test respond adequately in both hydrophilic (including aqueous) and organic solvents. However, the quantitative data obtained in our studies are very similar applying both tests, as shown in Figure 5.
9-In lines 244-250; Give reasons for the differences in results obtained if compared to old reported method Elouaddari et al. [20],
R: Possibly, the quantitative differences between our data and the corresponding to the study of Elouaddari et al. [21] can be due to the different methodological approach considering different organ zones or parts and, mainly by different solvent extraction methods. We have added this consideration in the new version (lines 264-266).
10-Write full name for abbreviation AA in table 12.
R: Thank you for your comment. We already have added the full name for AA in the table.
11-In line 339, the present study, all the methanolic extracts of C. mixtus gave very good results against T. rubrum. Did the authors tested methanol as a blank solvent to estimate that the activity is related to the plant and not the methanol itself? This is very important especially old papers confirmed that the plant extract has no inhibitory activity as stated in ref 37.
R: Thank you very much for your comment. About the methanol we did not use it in MIC, we dissolved plant extracts only in DMSO. The methanol was used just for the extraction of the biomolecules from the plants, and it was evaporated afterwards. For this reason, it is not necessary to test the methanol as a blank solvent. While we tested the DMSO as a blank solvent.
12-Derivatization reagents should be stated in the abstract. This point is of interest for readers.
R: Thank you for your suggestion. We have added derivatization reagents in Abstract, lines 29-30.
Round 2
Reviewer 1 Report
After revision, the manuscript was improved.
Reviewer 2 Report
the authors did all required recommendations. I appreciate their efforts. the paper could be published in the current form.
Greetings